# Astragaloside IV Inhibits the Proliferation of Human Uterine Leiomyomas by Targeting IDO1

**DOI:** 10.3390/cancers14184424

**Published:** 2022-09-12

**Authors:** Tiantian Qiu, Donghua Li, Yu Liu, Hui Ren, Xuan Yang, Wenting Luo

**Affiliations:** School of Traditional Chinese Medicine, Capital Medical University, Beijing 100069, China

**Keywords:** astragaloside IV, uterine leiomyomas, indoleamine-2, 3-dioxygenase-1, apoptosis

## Abstract

**Simple Summary:**

Immunotherapy is increasingly becoming a success strategy for oncology treatment. Indoleamine-2,3-dioxygenase1 (IDO1) is a tryptophan-degrading enzyme involved in immunological escape mechanisms, which is considered as a potential target for tumor therapy. However, the clinical efficacy of IDO1 inhibitors is not promising. Therefore, there is an urgent to investigate the mechanism between chemical drugs with antitumor effects and IDO1-mediated immunosuppression. The Chinese medicine AS-IV exerts antitumor effects with many advantages, including fewer toxic side effects and immunomodulatory effects. We noted the lack of studies of AS-IV on benign tumors. Therefore, our study demonstrates the Inhibitory effect of AS-IV on ULMs and elucidates the underlying mechanism.

**Abstract:**

Astragaloside IV (AS-IV) is a chemical found in traditional Chinese medicine called *Astragalus membranaceus* (Fisch.) Bunge that has antitumor properties. However, the roles and mechanisms of AS-IV in uterine leiomyomas (ULMs) are unclear. The immunosuppressive enzyme indoleamine-2,3-dioxygenase-1 (IDO1) is involved in tumor formation. IDO1 is a new and reliable prognostic indicator for several cancers. In this work, AS-IV was applied to ULM cells in various concentrations. CCK-8, immunofluorescence, and flow cytometry were used to examine the proliferation and apoptosis of ULM cells caused by AS-IV. After lentiviral vector transduction with IDO1 short hairpin RNA (shRNA), the knockdown and overexpression of IDO1 were stable in ULM cells. To verify the antitumor effect of AS-IV in vivo, we established a rat model of uterine leiomyoma. HE staining, Masson staining, and transmission electron microscopy were used to observe pathological changes in the uterus, and the levels of serum sex hormones were measured by radio immune assay (RIA). The levels of CD3+T, CD4+T, and CD25+ Foxp3+Treg in rat peripheral blood were detected by flow cytometry. Western blotting and immunohistochemistry were used to examine protein expression. We found that AS-IV dramatically increased the apoptotic rate of ULM cells and reduced viability in a time- and dosage-dependent manner. After sh-IDO1 lentiviral transfection, we discovered that knocking down IDO1 reversed the effects of AS-IV on ULM cell proliferation and autophagy. We also found that AS-IV can effectively inhibit the growth of ULMs in vivo. AS-IV may promote apoptosis and autophagy in ULMs by activating PTEN/PI3K/AKT signaling through inhibition of IDO1. These findings imply that AS-IV exerts antifibroid effects, and the underlying mechanism may be IDO1, which is involved in proliferation, apoptosis, and autophagy.

## 1. Introduction

Uterine leiomyomas (ULMs), the most frequent type of gynecologic tumor, are benign smooth muscle tumors that occur in the myometrium. The pathogenesis of ULMs is unknown, and current studies suggest that both estrogen and progesterone promote the development of leiomyomas [1]. Common symptoms include pelvic pain and pressure, low back pain, dysmenorrhea, infertility, excessive and prolonged menstruation, anemia, constipation, urinary incontinence, sexual dysfunction, recurrent miscarriages, and fibroids compressing neighboring pelvic organs [2]. Modern medical treatment includes pharmacological treatments, such as gonadotropin-releasing hormone (GnRH) agonists, myomectomy, hysterectomy, uterine artery embolization, and MR-guided focused ultrasound [3]. Clinically, ULMs are the most common reasons for hysterectomy. Complications of hormonal therapy and surgical resection severely affect the quality of life of patients. Therefore, a safe and effective alternative treatment is needed to address these existing drawbacks. Herbal natural chemicals may be useful in decreasing tumor progression, alleviating surgery-related discomfort, enhancing immunological function, and preventing problems associated with other therapeutic techniques [4].

*Astragalus membranaceus* (Fisch.) Bunge is widely used in the treatment of viral and bacterial infections, inflammatory conditions, and cancer [5]. Astragaloside IV (AS-IV), a tetracyclic triterpenoid saponin of the lanolin alcohol type, is one of the principal constituents in astragalus aqueous extract. It is chemically known as 3-O-β-D-xylopyranosyl-6-O-β-D-glucopyranosyl-cycloastragenol(C_14_H_68_O_14_). AS-IV is recognized as a quality control indicator for Astragalus in Chinese and European pharmacopeias [6]. Since ancient times, it has been widely used in China with no significant hepatotoxic or nephrotoxic consequences.AS-IV has been shown to exert significant anticancer effects or increase sensitivity to other drugs when used alone or in combination with other therapeutic modalities [7,8]. The mechanisms of AS-IV include induction of apoptosis in tumor cells [9], inhibition of proliferation [10], inhibition of metastasis [11], inhibition of angiogenesis [12], promotion of autophagy [13], and enhancement of immunity [14].

Immune checkpoint blockade leads to unprecedented responses in various cancers. An alternative approach to unleash the antitumor immune response is to target the inhibition of immune metabolic pathways, such as the indoleamine 2,3-dioxygenase1 (IDO1) pathway. IDO1 overexpression has been associated with cancer progression in an increasing number of studies. Furthermore, elevated IDO1 levels have been linked to a worse prognosis in individuals with ovarian cancer, hepatocellular carcinoma, invasive cervical cancer, and non-small cell lung cancer [15,16,17,18,19]. IDO1 has been demonstrated to serve as a novel and reliable prognostic indicator for many types of cancer. Generally, IDO1 suppresses T-cell function through induced tryptophan starvation, leading to tumor development [20,21]. However, Thaker [22] discovered that IDO1 directly regulates colon cancer proliferation and progression, independent of its immunoregulatory actions. In our previous study, we found that curdione inhibited uterine leiomyosarcoma cell proliferation by targeting IDO1 to induce G2/M phase block, apoptosis, and autophagy [23]. As a result, we hypothesized that AS-IV may affect ULMs by inhibiting IDO1.

## 2. Materials and Methods

### 2.1. Reagents and Materials

Astragaloside IV (Solarbio, Science&Technology, Beijing, China; cat:SA8640; HPLC ≥ 98%). Dulbecco’s Modified Eagle Medium (DMEM; cat:12320032), fetal bovine serum (FBS; cat:26170043), and penicillin–streptomycin (PS; cat:10378016) were purchased from Gibco (Waltham, MA, USA). Dimethyl sulfoxide (DMSO; cat:D4540) was purchased from Sigma-Aldrich (St. Louis, MO, USA). Estradiol benzoate injection (Jinke, Sichuan, China); progesterone injection (Xianju, Zhejiang, China); mifepristone tablets (Resources Zizhu, Beijing, China). Anti-IDO1(cat: ab211017), anti-PI3KCA (cat: ab40776), anti-AKT1(cat: ab179463), anti-LC-3(cat: ab48394), anti-FOXO3A (cat: ab154786), anti-PCNA (cat: ab92552), and anti-βactin (cat:ab8226) antibodies were purchased from Abcam (Cambridge, UK). Anti-caspase-9 (cat:10380-1-AP), anti-FADD (cat:14906-1-AP), and anti-PTEN (cat:22034-1-AP) antibodies were obtained from Proteintech (Wuhan, China). PerCP/Cyanine5.5 anti-rat CD3 (cat: 201417), Alexa Fluor^®^ 488 anti-rat CD4 (cat: 201511), and APC anti-rat CD25 (cat: 202113) were purchased from Biolegend (San Diego, CA, USA). FOXP3 Monoclonal Antibody (150D/E4)-PE (eBioscience, San Diego, CA, USA; cat:12477441).

### 2.2. Tissue Collection

Uterine leiomyomas tissues were obtained from regular menstruating Chinese women who underwent myomectomy at Beijing Maternity Hospital, Capital Medical University. The Institutional Review Board authorized the study protocol for collection of surgical specimens. Before surgery, patients provided informed consent for the use of leiomyoma tissue in this study. Patients did not receive hormonal therapy for ≥6 months before surgery. Surgical specimens were diagnosed postoperatively by pathology as uterine fibroids.

### 2.3. Cell Culture

ULM-1 cells and ULM-2 cells were obtained from human uterine leiomyomas tissues of patients. ULM-2 cells were provided by Procell Life Science & Technology Co., Ltd. (Wuhan, China; cat: CP-H151). The culture process of ULM-1 cells was as follows. The tissues were washed three times with PBS containing 1% penicillin-streptomycin. The center parts of fresh leiomyoma tissues were obtained after carefully removing pseudo-capsules and fibrous septa materials, then cut into 1 mm^3^ pieces, put into 0.2% type I collagenase, and digested at 37 °C for 3–6 h. The cell suspension was filtered with a 70 µm cell sieve, and the leiomyoma cells were collected by centrifugation at 1250 rpm for 5 min and washed 3 times with PBS. The isolated cells were inoculated at a density of approximately 1 × 10^6^ cells/dish in a 10 cm^2^ culture dish with DMEM medium containing 10% FBS and 1% penicillin-streptomycin and incubated in a sterile environment with 5% CO^2^ at 37 °C for 24 h.

### 2.4. Cell-Counting Kit (CCK)-8 Assay

A CCK-8 kit (Kumamoto, Japan; cat: CK04) was used to test the vitality of suitably treated cells. In brief, the 100 µL cell suspension was cultured at a density of 1 × 10^4^ and grown for 24 h in 96-well plates with AS-IV (0, 1, 10, 50, 100, 200, 300, and 400 µM) for 24 h, 48 h, and 72 h. Then, 10 µL CCK-8 solution was added and incubated for 2 h in a sterile environment with 5% CO_2_ at 37 °C. The optical density (OD) values at 450 nm were then measured using a microplate spectrophotometer (Molecular Devices, Sunnyvale, CA, USA).

### 2.5. Immunofluorescence

The ULM cells were fixed with 4% paraformaldehyde for 20 min, permeabilized with 0.3% TritonX-100 for 25 min, and blocked with 10% goat serum for 30 min in 24-well plates. Then, primary antibody PCNA was added and incubated overnight at 4 °C. Secondary antibody (FITC/TRITC-conjugated goat anti-rabbit IgG) was added and reacted for 1 h. The images were observed under a laser confocal microscope (Leica TCS SP8).

### 2.6. Flow Cytometry Analysis

The apoptotic rate of ULM cells was detected using an annexin V-FITC/PI apoptosis assay Kit (Beyotime, Shanghai, China; cat: C1062M). After a sufficient number of cells was collected, 5 µL annexin V-FITC and 10 µL propidium iodide (PI) staining solution were added and incubated for 20 min under low light. Flow cytometry (BD LSR Fortessa) was used to determine the percentages of apoptotic cells. 

Peripheral blood was collected from rats, and lymphocytes were isolated. CD3, CD4, and CD25 antibodies were added and incubated for 15 min and protected from light. After fixing and breaking the membrane, the Foxp3 antibody was added and incubated for 30 min at room temperature. The ratio of Treg cells was detected by flow cytometry (CytoFlex S, Brea, CA, USA).

### 2.7. Western Blotting

Protein levels of caspase-3, caspase-9, FADD, Beclin-1, LC3, FOXO3A, p62, PI3KCA, AKT1, IDO1, FOX01, and PTEN were detected by Western blotting analysis. Total proteins were extracted from ULM cells, and protein concentrations were standardized by BCA protein assay kit (NCM, Suzhou, China; cat: WB6501). The equal proteins were separated by 10% to 15% SDS-PAGE and transferred onto polyvinylidene difluoride (PVDF) (Millipore, MA, USA; cat: IPVH00010). After blocking with NcmBlot blocking buffer (NCM, Suzhou, China; cat: P30500), the membranes were incubated overnight at 4 °C with the following primary antibodies: caspase-9, FADD, LC3, FOXO3A, PI3KCA, AKT1, IDO1, PTEN, and β-actin. After washing, the membranes and secondary antibodies were incubated for 1 h. The target protein was exposed on an imaging system (Bio-Rad, Hercules, CA, USA) using the and enhanced chemiluminescence (ECL) detection kit (Genview, Houston, TX, USA; cat: GE2301). Western blotting trials were repeated with at least three independent specimens, and the reported results are representative.

### 2.8. Real-Time Quantitative PCR

Total RNA was extracted from uterine leiomyoma cell samples using a FastPure Cell/Tissue total RNA isolation kit V2 (Vazyme, Nanjing, China; cat: RC112), and the concentration was measured. HiScript III RT SuperMix for qPCR (+gDNA wiper) was used to synthesize cDNA (Vazyme, Nanjing, China; cat: Q711). RT-PCR was performed using ChamQ Universal SYBRqPCR master mix (Vazyme, Nanjing, China; cat: RC112) on a CFX96 real-time system (Bio-Rad, USA).

The primers were designed as follows:

IDO1: F-TGGCCAGCTTCGAGAAAGAG and R-GATAGCTGGGGGTTGCCTTT;

GAPDH: F-GGAGCGAGATCCCTCCAAAAT and R-GGCTGTTGTCATACTTCTCATGG.

### 2.9. Lentiviral Vector-Mediated IDO1 Overexpression and Knockdown in ULMs

Overexpression of lentiviral vector HBLV-IDO1-3xflag-ZsGreen-PURO (titer: 1 × 10^8^ TU/mL, lot:LV54041326), knockdown lentiviral vector HBLV-h-IDO1 shRNA1-ZsGreen-PURO (titer: 1 × 10^8^ TU/mL, lot:LV54042967), HBLV-h-IDO1 shRNA2-ZsGreen-PURO (titer: 3 × 10^8^ TU/mL, lot:LV54042964), HBLV-h-IDO1 shRNA3-ZsGreen-PURO (titer: 1 × 10^8^ TU/mL, lot: LV54042963), empty virus control HBLV-ZsGreen-PURO NC (titer: 3 × 10^8^ TU/mL, lot: LV54042962), and puromycin were purchased from Hanbio (Shanghai, China).

shRNA sequences of IDO1 include:
NC-shRNA:5’-TTCTCCGAACGTGTCACGTAA-3’,sh1-IDO1:5’-TGCTAAACATCTGCCTGATCTCATA-3’,sh2-IDO1:5’CCTACTGTATTCAAGGCAATGCAAA-3’,sh3-IDO1: 5’-CCCTGAGGAGCTACCATCTGCAAAT-3’.

ULM cells (2 × 10^5^/well) were inoculated into six-well cell culture plates and transfected with a lentiviral vector in 1/2 volume of media. After transfection for 72 h, transfection efficiency was observed under a fluorescence microscope, and the stably transfected cell lines were selected with 1 µg/mL puromycin. The successfully transfected uterine leiomyoma cells were passaged and cultured in a normal medium. Then, qRT-PCR was used to confirm the long-term transfection efficacy.

### 2.10. Rat Uterine Fibroid Model

Forty non-pregnant adult SPF-grade Sprague–Dawley rats (2–3 months old, female; weight, 200 ± 10 g) were provided by the Department of Laboratory Animals, Capital Medical University (Beijing, China). All animal experiments were approved by the Animal Ethics Committee of Capital Medical University. Rats were randomly divided into 5 groups: normal group (Control), model group (Model), positive group (MF), AS-IV high-dose group (AH), AS-IV low-dose group (AL); *n* = 8. Rats were acclimatized for one week. The model group, MF group, AH group, and AL group were intraperitoneally injected with estradiol benzoate once every other day (0.5 mg/kg) and progesterone once a week (4 mg/kg), and the two hormones were injected simultaneously in the last 5 days for 10 weeks. The control group received an intraperitoneal injection of the same volume of physiological saline daily. From the 6th week of model induction, the AH and AL groups were gavaged with AS-IV (80 mg/kg, 40 mg/kg), and the MF group was gavaged with mifepristone (2.9 mg/kg) daily. At the end of the experiment, the rats were fasted for 12 h, anesthetized, and executed, and uterine tissues were taken for pathological observation and Western blotting tests.

### 2.11. Histopathological Examination of the Uterus

Rat uterine slices were dewaxed in xylene and washed in an ethanol gradient. Then, the slices were stained with hematoxylin for 10 min, eosin for 3 min, and lixin for 10 min. After dehydration and transparency were achieved with ethanol and xylene, the film was sealed with a drop of neutral resin. The images were observed and photographed under a light microscope (Leica DM60008, Kyoto, Japan) (×200).

### 2.12. Transmission Electron Microscopy

Rat uterine tissues were fixed, dehydrated, permeabilized, and embedded. Then, the tissue was sectioned (thickness 80–100 nm). The sections were stained with 2% uranyl acetate saturated in water and lead citrate for 15 min and dried at room temperature overnight. The images were observed and photographed using an electron microscope (JEM-2100, JEOL. Tokyo, Japan).

### 2.13. Radio Immune Assay (RIA)

According to the manufacturer’s instructions, an iodine (^125^I)-estradiol radioimmunoassay kit (Xiehe Medical Technology Co., Ltd., Tianjin, China), an iodine (^125^I)-human luteinizing hormone radioimmunoassay kit (Xiehe Medical Technology Co., Ltd., Tianjin, China), an iodine (^125^I)-human follicle stimulating hormone radioimmunoassay kit (Xiehe Medical Technology Co., Ltd., Tianjin, China), and an iodine (^125^I)-progesterone radioimmunoassay kit (Xiehe Medical Technology Co., Ltd., Tianjin, China) were used to detect changes in the levels of estradiol(E2), follicle-stimulating hormone(FSH), luteinizing hormone (LH), and luteinizing hormone (P) in rat serum. The values were measured by a γ- type immune counter (GC-2010, ZONKIA, Hefei, China).

### 2.14. Immunohistochemistry

Immunohistochemistry was used to detect the expression of IDO1, PI3KCA, AKT1, PTEN, caspase-9, FADD, LC-3, and FOXO3A proteins in uterine tissues of rats. The images were observed using a light microscope (Leica DM60008, Kyoto, Japan). Mean optical density values (OD) were analyzed using Image-Pro Plus 6.0 (Media Cybernetics, Inc., Rockville, MD, USA).

### 2.15. Statistical Analysis

The results are presented as mean ± standard deviation (SD). Statistical analysis was carried out by one-way analysis of variance (ANOVA) using SPSS version 19.0 software (SPSS Inc., Chicago, IL, USA) and GraphPad Prism 7.0 software. Statistical significance was defined as *p* < 0.05.

## 3. Results

### 3.1. AS-IV Inhibited the Proliferation of ULM Cells

A CCK-8 assay was used to determine the antiproliferative effect of AS-IV on ULM cells. As demonstrated in Figure 1A–C, AS-IV dramatically decreased ULM cell viability in a time- and dosage-dependent manner. The IC50 (half-maximal inhibitory concentration) for ULM-1 cells was 205.9 µM (Figure 1E), and that for ULM-2 cells was 215.0 µM (Figure 1F). To ensure that most of the cells were in good condition and to verify the feasibility of subsequent experiments, we selected an IC50 (100 µM) of less than 1/2 as the optimal concentration. Figure 1D shows the chemical structure of AS-IV.

### 3.2. AS-IV Induced Apoptosis in ULM Cells

Annexin V-FITC/PI assays were used to detect the impact of different doses of AS-IV on apoptosis in ULM-1 and ULM-2 cells. We found that AS-IV significantly increased the apoptosis rates of ULM cells in a dosage-dependent manner (Figure 2A–D).

Through cell proliferation and apoptosis detection, we found that AS-IV had similar effects on ULM-1 cells and ULM-2 cells, whereas ULM-1 cells exhibited a better growth state. Therefore, we selected ULM-1 cells for subsequent experiments.

### 3.3. IDO1 Involved in the Suppressive Effect of AS-IV in ULM Cells

We induced stable IDO1 knockdown and overexpression in ULM cells. The ZsGreen positive cell ratio was used to indicate the efficiency of target gene transfection in ULM cells (Figure 3A). The efficiency of lentiviral vector transfection of ULM-NC-IDO1 (Transfection vector control group), ULM-OE-IDO1 (IDO1 gene overexpression group), ULM-sh1-IDO1 (IDO1 gene knockdown group1), ULM-sh2-IDO1 (IDO1 gene knockdown group2), and ULM-sh3-IDO1 (IDO1 gene knockdown group3) after 72 h in ULM cells was more than 60%. In addition, RT-PCR (Figure 3C) results suggest that lentivirus-mediated transfection is effective and stable. To study the effect of IDO1 on the inhibition of ULM cell proliferation by AS-IV, we used CCK-8 and immunofluorescence assays. Compared with the control group, the cell viability in the ULM-sh1-IDO1, ULM-sh2-IDO1, and ULM-sh3-IDO1 groups was considerably decreased (Figure 3D). However, the cell viability was dramatically increased in the ULM-sh-IDO1+A-IV group and decreased in the ULM-OE-IDO1+A-IV group compared with the AS-IV-only group (Figure 3B). Immunofluorescence produced similar results. The PCNA protein expression was considerably increased in the ULM-sh-IDO1+A-IV group and decreased in the ULM-OE-IDO1+A-IV group compared with the AS-IV-only group (Figure 4A,B). These findings suggest that IDO1 may be involved in the inhibitory effect of AS-IV on ULM cells.

### 3.4. IDO1 Involved in the Apoptosis Induced by AS-IV in ULM Cells

To elucidate the potential mechanism by which IDO1 is involved in the inhibitory effect of AS-IV in ULM cells, we performed a flow cytometry assay. Compared with the AS-IV-only group, the ULM-sh-IDO1+A-IV group exhibited a significant decrease in the apoptosis rate (Figure 5). These results indicate that IDO1 may be involved in the inhibitory effect of AS-IV in ULM cells by modulating apoptosis.

### 3.5. Antitumor Effects of AS-IV In Vivo

To verify the antitumor effect of AS-IV in vivo, we established a rat model of uterine leiomyoma. According to the general observation of uterine tissue (Figure 6), in the control group, the “Y”-shaped uterine morphology and structure were normal, and the left and right branches presented equal thickness values. In the model group, the uterus was dark red with multiple nodules, and uneven surface, uneven thickness of the left and right branches, and a large area of congestion and edema. In the MF, AH, and AL groups, the uterus was nearly normal, with occasional tiny nodules. We measured rat body weight (Figure 7A), uterine weight (Figure 7B), and uterine transverse diameter (Figure 7C). Compared to the model group, the uterine coefficient (uterus weight/rat body weight) and uterine transverse diameter were significantly reduced in the MF, AH, and AL groups. We observed and analyzed pathological structures of the uterus (Figure 8). In the control group, the uterine smooth muscle cells were neatly arranged, with regular morphology and clear and intact nuclei and cytoplasm. In the model group, the uterine smooth muscle cells were hypertrophied and shortened, with obvious thickening of the muscle layer and large fibrous connective tissue between the muscle bundles. In the MF, AH, and AL groups, the uterine smooth muscle cells were nearly normal in structure. The serum sex hormone levels of rats were measured by radio immune assay (RIA). Compared with the model group, the concentrations of FSH, LH, E2, and P were significantly lower in the AH and AL groups (Figure 9A–D). The above results indicate that AS-IV can effectively inhibit the growth of uterine fibroids in vivo.

### 3.6. Mechanism of Inhibition of ULM Growth by AS-IV In Vivo

To further clarify the mechanism of AS-IV in inhibiting the growth of ULMs in vivo, we performed Western blotting, immunohistochemistry, and flow cytometry. The immunohistochemistry results show that the expressions of IDO1, PI3KCA, and AKT1 were significantly reduced in the AH and AL groups, and the expressions of PTEN, caspase-9, FADD, LC-3, and FOXO3A were significantly increased compared to the model group (Figure 10A,B). Similarly, the above results were further confirmed by Western blotting (Figure 11A,B). The flow cytometry results show that CD3+T and CD4+T cells increased significantly, and CD25+FOXp3+Treg cells decreased significantly in the AH and AL groups compared to the model group (Figure 12A–D). These findings suggest that AS-IV may promote apoptosis and autophagy in ULMs by activating PTEN/PI3K/AKT signaling through inhibition of IDO1.

## 4. Discussion

In our study, we report that AS-IV inhibits the growth of ULMs in vivo and in vitro. AS-IV decreased ULM cell viability in a time- and dosage-dependent manner and effectively inhibited proliferation. AS-IV showed excellent antitumor effects in vivo. Apoptosis and autophagy are two types of programmed cells, and deregulation is an important mechanism of tumorigenesis [24]. Apoptosis is primarily triggered by the mitochondrial pathway, the endoplasmic reticulum pathway, and the death receptor pathway [25]. In our study, AS-IV induced apoptosis in a dosage-dependent manner in ULM cells, which was associated with increased levels of caspase-9 and FADD (Fas-associated protein with death domain). The results suggest that AS-IV may mediate apoptosis through the mitochondrial pathway and the death receptor pathway. Autophagy is usually considered a self-protection mechanism of cells and plays a dual role in tumor formation. We found that AS-IV promoted autophagy occurrence in ULM rats by upregulating LC3 and FOXO3A levels. Nevertheless, whether the occurrence of autophagic inhibits ULMs growth needs to be further explored.

The PTEN/PI3K/AKT signaling plays an important role in the regulation of cell proliferation, growth, apoptosis, and autophagy [26]. The tumor suppressor phosphatase and tensin homolog (PTEN) is an endogenous inhibitor of PI3K signaling, which is usually mutated in tumors [27]. The silencing of PTEN leads to the activation of the PI3K/AKT pathway, which affects apoptosis, autophagy, EMT (epithelial–mesenchymal transition), and other processes of tumor cells [28]. The PI3K/AKT signaling pathway has been identified to be highly activated in ULM cells [29]. The mechanism of increased activation is unknown and may involve modifications of receptor tyrosine kinase, PI3K, PTEN, and AKT, and tyrosine kinase receptors may regulated IDO1 expression in tumors through PI3K-AKT signaling [30,31]. Furthermore, Kumar [32] found that IDO1 and kynurenine pathway (KP) metabolites directly promote tumorigenesis by activating PI3K/AKT signaling. PTEN/PI3K/AKT signaling is considered a potential target for tumor therapy, and many inhibitors have been clinically developed for cancer chemotherapy. Jia [33] found that AS-IV can inhibit mitochondria-mediated cardiomyocyte apoptosis through activation of the PI3K/AKT signaling, thereby attenuating doxorubicin-induced cardiotoxicity in the antitumor process. In our study, Western blotting and immunohistochemical results showed that AS-IV significantly downregulated the protein expression of IDO1, PI3KCA, and AKT1 while upregulating the protein level of PTEN. Therefore, we hypothesized that AS-IV may promote apoptosis and autophagic processes in ULMs through PTEN/PI3K/AKT signaling, which involves IDO1.

Traditional tumor treatments, including surgical resection and chemotherapy, have disadvantages in many aspects [15,34,35]. Indoleamine-2,3-dioxygenase (IDO) is a tryptophan-degrading enzyme involved in immunological escape mechanisms. Previous studies revealed that inhibition of IDO1 restores T-cell activity and improves the ability to kill tumor cells [36,37]. Regulatory T cells (Treg) are the most important cells in the tumor microenvironment. IDO1 can directly cause T-cell suppression and induce Treg activation by depleting tryptophan and producing kynurenine [21]. Zhang [38] found that AS-IV significantly increased the ratio of Tregs and decreased the ratio of CTLs (cytotoxic T lymphocytes) by inhibiting IDO, thereby inhibiting immune escape from tumors and suppressing the progression of lung cancer. In our study, AS-IV was found to inhibit proliferation and downregulate IDO1 expression in ULMs. In vivo experiments further showed that AS-IV could inhibit the growth of ULMs by modulating Treg. These results suggest that AS-IV may exert antitumor and immune-enhancing effects by inhibiting Treg activation through suppression of IDO1 expression. Furthermore, sh-IDO1 lentiviral transfection was used to knock down or increase IDO1 expression in ULM cells. Surprisingly, blocking IDO1 significantly attenuated the inhibitory effect of AS-IV on ULM cells, which also reversed AS-IV-induced apoptosis. These data show that IDO1 was involved in the inhibitory effect of AS-IV on ULMs. However, AS-IV has been shown to downregulate IDO1 expression and inhibit Treg activity, which may restore tumor immunomodulatory effects and inhibit ULM growth. Repeated immunosuppression may promote expansion of Treg subsets [39]. Long-term monitoring of intratumoral T-cell populations has shown reduced long-term survival and reduced antitumor cytotoxicity after repeated immunotherapy [40]. Therefore, the enhanced inhibitory activity and the intensified Treg rebound after AS-IV and IDO1 knockdown on ULM cells may be responsible for the loss of the antitumor effect of AS-IV.

In summary, our study demonstrates the antifibroid effects of AS-IV in vitro and in vivo and elucidates the underlying mechanism. AS-IV induces apoptosis and autophagy and inhibits proliferation of ULMs through inhibition of IDO1, which involves the activation of PTEN/PI3K/AKT signaling and regulation of Treg. These findings highlight the importance of studying IDO1 inhibition in ULMs and provides new perspectives with respect to the potential of AS-IV for anticancer therapy.

## Figures and Tables

**Figure 1 cancers-14-04424-f001:**
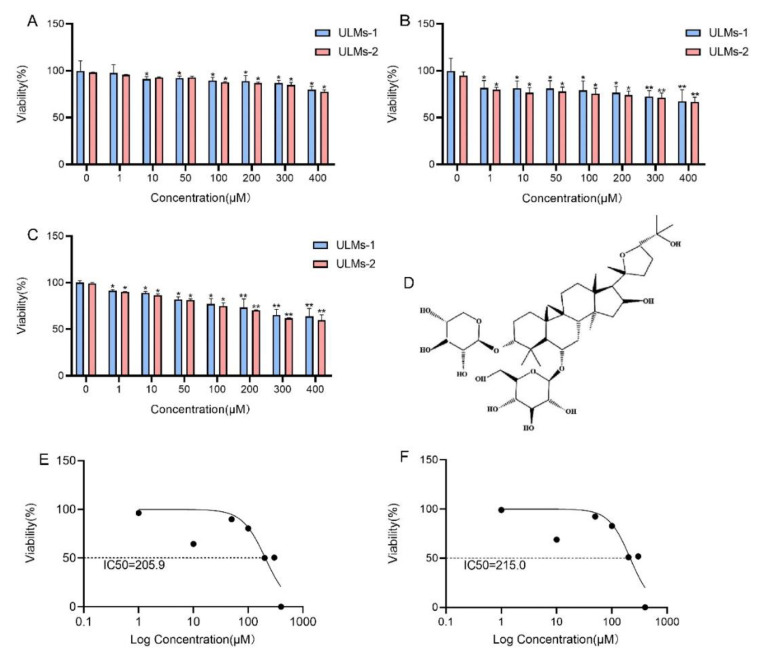
Effect of AS-IV on the proliferation of ULM cells. ULM-1 cells and ULM-2 cells were treated in various concentrations of AS-IV (0, 1, 10, 50, 100, 200, 300, and 400 µM), and changes in cell viability were observed at (**A**) 24 h, (**B**) 48 h, and (**C**) 72 h. (**D**) Chemical structure of AS-IV. IC50 values of ULM-1 (**E**) and ULM-2 cells (**F**) were calculated using GraphPad prism8 and non-linear regression. The data are expressed as the mean ± SD. (* *p* < 0.05, ** *p* < 0.01 compared with the control group).

**Figure 2 cancers-14-04424-f002:**
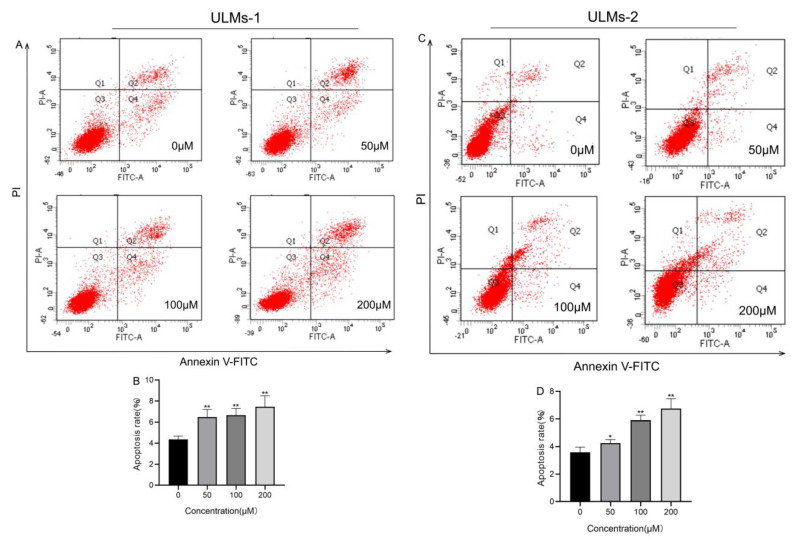
AS−IV induces apoptosis in ULM cells. ULM−1 and ULM−2 cells were treated with various concentrations of AS−IV (0 µM, 50 µM, 100 µM, and 200 µM) for 72 h. Determination of the apoptotic rate of ULM−1 cells (**A**,**B**) and ULM−2 cells (**C**,**D**) using an annexin V−FITC/PI apoptosis assay kit. The data are expressed as the mean ± SD. (***
*p* < 0.05, ****
*p* < 0.01 compared with the control group).

**Figure 3 cancers-14-04424-f003:**
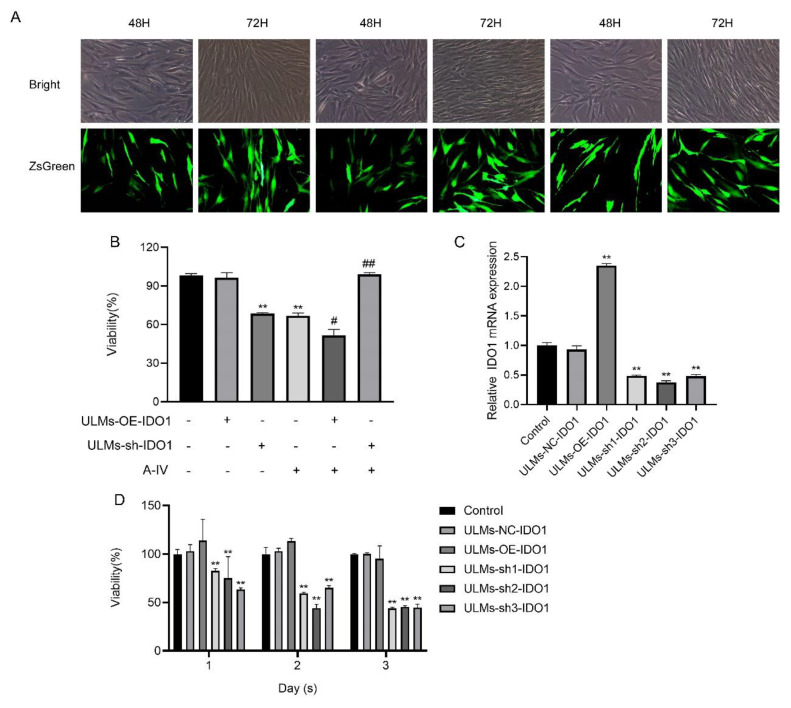
IDO1 is involved in the proliferation of AS−IV on ULM cells. (**A**) ULM cells were transfected with shRNA lentiviral vector for 48 h and 72 h. Then, the cell growth state was observed with a light microscope (Bright), and the expression of green fluorescence was observed with a fluorescence microscope (×20 magnification). (**B**) CCK−8 analysis shows the effect of proliferation in ULM cells after transfection with shRNA−IDO1 by AS−IV. (**C**) Real-time quantitative PCR analysis indicates IDO1 mRNA expression in ULM cells. (**D**) A CCK−8 assay was used to detect the vitality of ULM cells after transfection with shRNA−IDO1 following 24 h, 48 h, and 72 h of culture. The data are presented as the mean ± SD (****
*p* < 0.01 compared with the control group; *#*
*p* < 0.05, *##*
*p* < 0.01 compared with the A-IV group).

**Figure 4 cancers-14-04424-f004:**
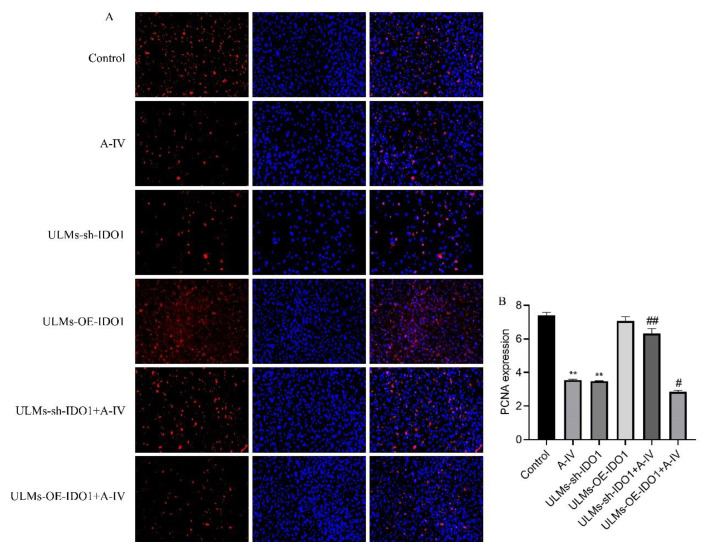
Immunofluorescence analysis shows the effect of proliferation in ULM cells after transfection with shRNA-IDO1 by AS-IV. (**A**) PCNA protein expression under fluorescence microscopy. (**B**) ImageJ was used to calculate positive fluorescence density. The data are presented as the mean ± SD (****
*p* < 0.01 compared with the control group; *#*
*p* < 0.05, *##*
*p* < 0.01 compared with the A-IV group).

**Figure 5 cancers-14-04424-f005:**
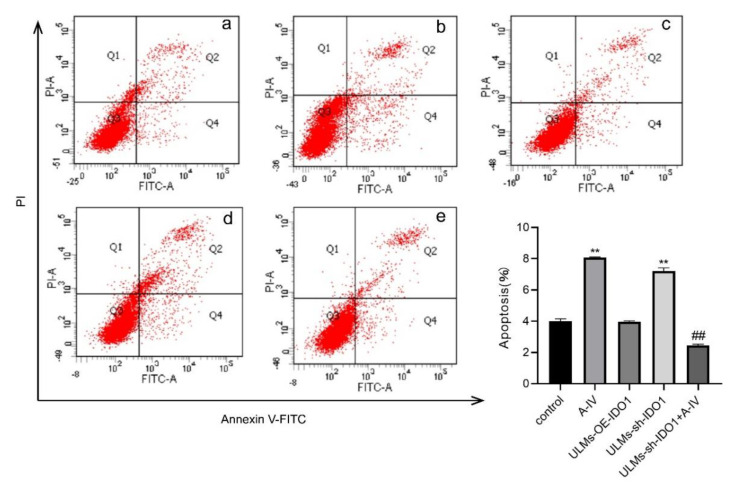
Detection of apoptosis rate in ULM cells by flow cytometry. (**a**) Control group; (**b**) A−IV group; (**c**) ULM−OE−IDO1 group; (**d**) ULM−sh−IDO1 group; (**e**) ULM−sh−IDO1 + A−IV group. Q1: Annexin V−/PI+, representing cellular debris or necrotic cells from other causes. Q2: Annexin V+/PI+, represents advanced apoptotic or necrotic cells. Q3: Annexin V−/PI−, represents normal living cells. Q4: Annexin V+/PI−, representing early apoptotic cells. The data are presented as the mean ± SD. (****
*p* < 0.01 compared with the control group; *##*
*p* < 0.01 compared with the A-IV group).

**Figure 6 cancers-14-04424-f006:**
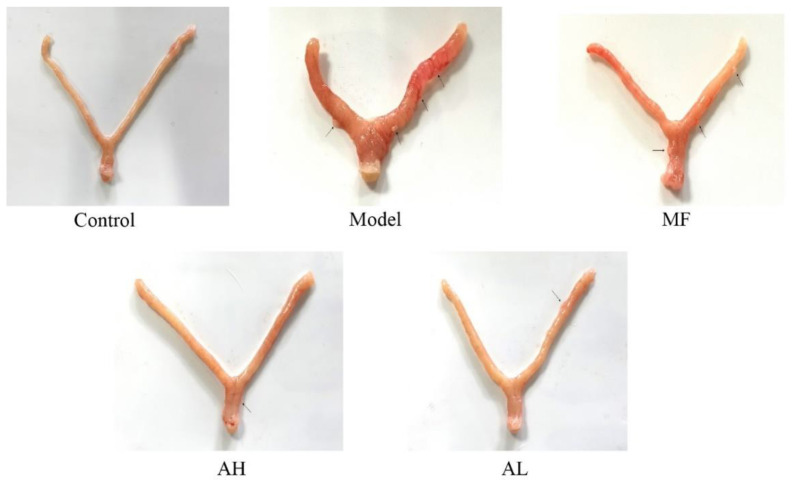
Representative images of rat uterine after AH and AL treatments. The arrow points out the location of the nodule in the uterine tissue.

**Figure 7 cancers-14-04424-f007:**
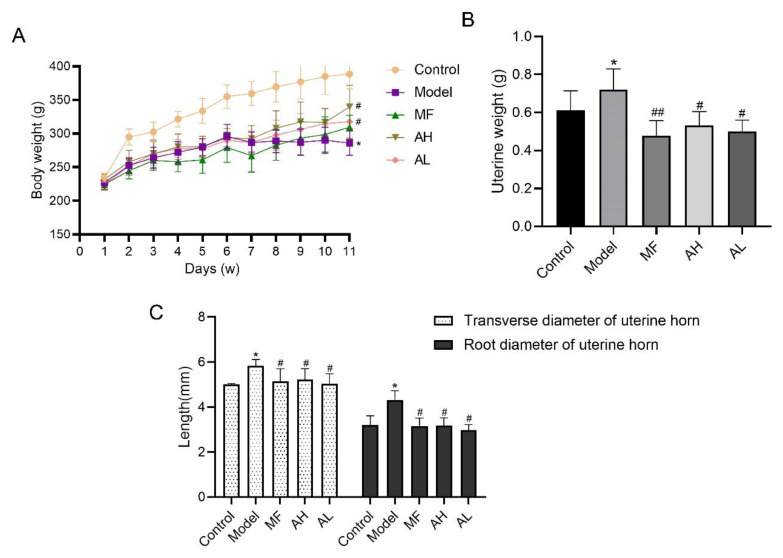
Rat body weight and uterine coefficient. (**A**) Weight changes in rats were monitored weekly. (**B**) Weight of uterus. (**C**) Transverse diameter of uterus. The data are presented as the mean ± SD (***
*p* < 0.05, compared with the control group; *#*
*p* < 0.05, *##*
*p* < 0.01 compared with the model group).

**Figure 8 cancers-14-04424-f008:**
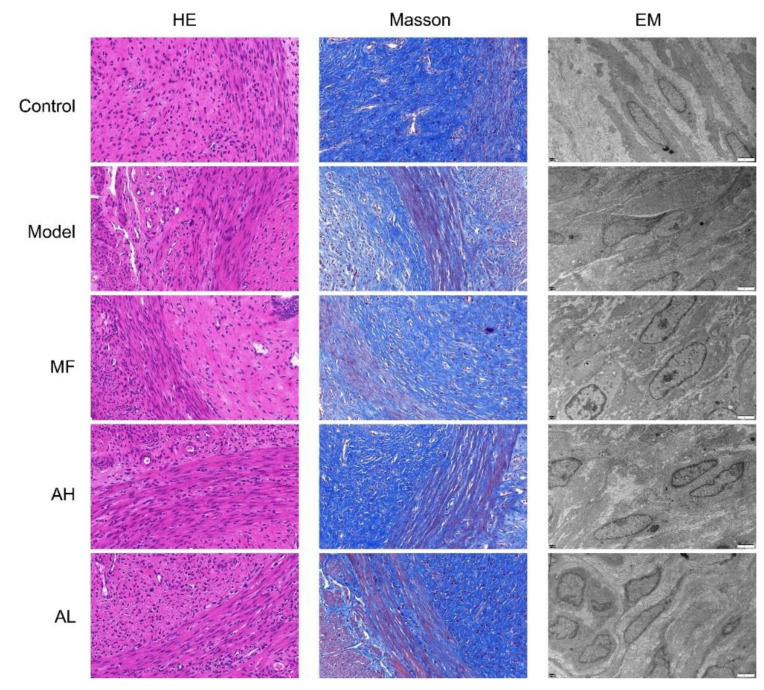
Pathological image of uterus smooth muscle tissue. HE: hematoxylin-eosin staining (×200), Masson (×200), EM: electron microscope (×8000).

**Figure 9 cancers-14-04424-f009:**
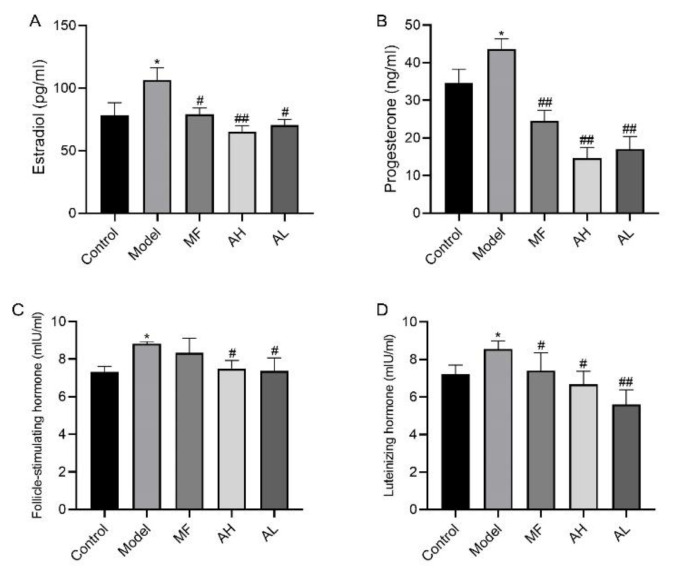
Determination of serum neutral hormone levels in rats by radio immune assay (RIA). (**A**) Estradiol. (**B**) Progesterone. (**C**) Follicle-stimulating hormone. (**D**) Luteinizing hormone. The data are presented as the mean ± SD (***
*p* < 0.05, compared with the control group; *#*
*p* < 0.05, *##*
*p* < 0.01 compared with the model group).

**Figure 10 cancers-14-04424-f010:**
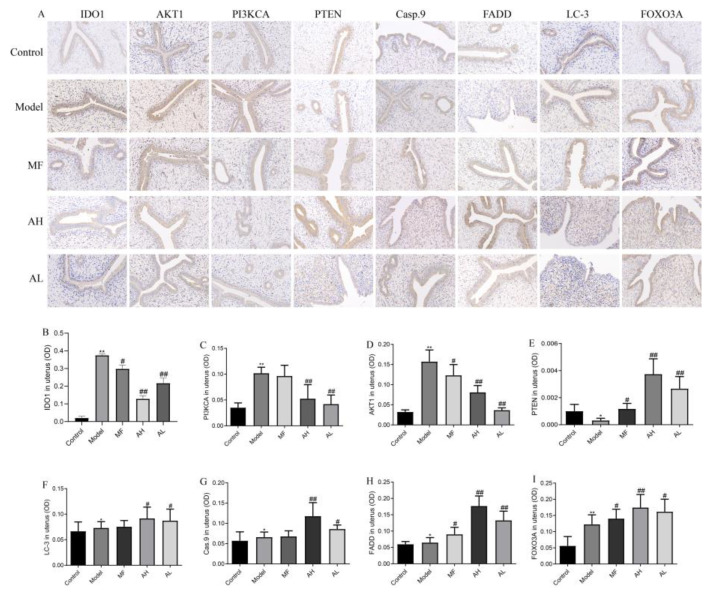
Protein expression of IDO1, PI3KCA, AKT1, PTEN, caspase-9, FADD, LC-3, and FOXO3A in rat uterine tissues by immunohistochemistry (×200). (**A**) Representative images of immunohistochemistry results. (**B**) Statistical results of IDO1 protein. (**C**) Statistical results of PI3KCA protein. (**D**) Statistical results of AKT1 protein. (**E**) Statistical results of PTEN protein. (**F**) Statistical results of LC-3 protein. (**G**) Statistical results of Casp.9 protein. (**H**) Statistical results of FADD protein. (**I**) Statistical results of FOXO3A protein. The data are presented as the mean ± SD (***
*p* < 0.05, ****
*p* < 0.01 compared with the control group; *# p* < 0.05, *## p* < 0.01 compared with the model group).

**Figure 11 cancers-14-04424-f011:**
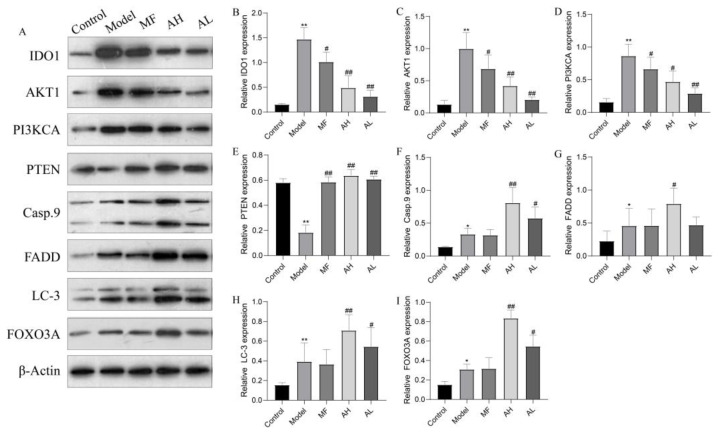
Protein expression of IDO1, PI3KCA, AKT1, PTEN, caspase-9, FADD, LC-3, and FOXO3A in rat uterine tissues by Western blotting. (**A**) Representative strip images of western blotting. (**B**) Statistical results of IDO1 protein. (**C**) Statistical results of AKT1 protein. (**D**) Statistical results of PI3KCA protein. (**E**) Statistical results of PTEN protein. (**F**) Statistical results of Casp.9 protein. (**G**) Statistical results of FADD protein. (**H**) Statistical results of LC-3 protein. (**I**) Statistical results of FOXO3A protein. The data are presented as the mean ± SD (** p* < 0.05, *** p* < 0.01 compared with the control group; *# p* < 0.05, *## p* < 0.01 compared with the model group).

**Figure 12 cancers-14-04424-f012:**
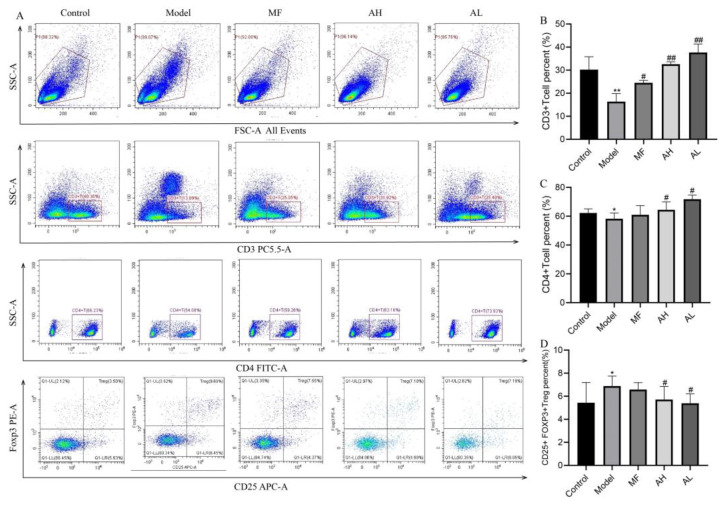
The levels of CD3+T, CD4+T, and CD25+ Foxp3+ Treg in rat peripheral blood were detected by flow cytometry. (**A**) After staining for CD3, CD4, and CD25/FOXP3 in T cells, detection was performed by flow cytometry. (**B**) Statistical results of CD3+Tcell. (**C**) Statistical results of CD4+Tcell. (**D**) Statistical results of CD25+Foxp3+Treg The data are presented as the mean ± SD (** p* < 0.05, *** p* < 0.01 compared with the control group; *# p* < 0.05, *## p* < 0.01 compared with the model group).

## Data Availability

All data generated or analyzed during this study are included in this published article.

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
