# Peer review of "Astragaloside IV Inhibits the Proliferation of Human Uterine Leiomyomas by Targeting IDO1"

_cancers, 2022, doi:10.3390/cancers14184424_

Round 1
Reviewer 1 Report
In this manuscript, Qiu et al. analyzed the mechanism of anti-tumor function of AS-IV in uterine leiomyomas and they found that it elicits its function through targeting IDO1. Although it is an original study, there are other closely related studies that should be discussed in the manuscript. One of them is a Phase 3 study for leiomyomas that includes Astragalus membranaceus TOGETHER WÄ°TH 14 OTHER Chinese medicines to use as Tumor-shrinking Decoction (TSD) (https://clinicaltrials.gov/ct2/show/study/NCT02189083). It would be better to discuss this in the manuscript. The other one is a 2014 paper analyzing the anti-tumor effect of AS-IV in lung cancer, and they found that AS-IV performs this action by interfering with IDO1 (https://pubmed.ncbi.nlm.nih.gov/24980548/). They should also discuss this paper in their manuscript.
In general, grammar should be carefully checked and corrected throughout the manuscript. Because there ara a lot of incorrect, incomplete, informal, or missing verb sentences. Here are the some examples:
- In the abstract, the sentence beginning as ‘ HE and MASSON staining…’ is incomplete and should be corrected.
- In materials- methods, under title 2.4 ‘ CCK-8. Assay’, the sentence ‘ then add 10 ul CCK-8 solution and incubate with cells ‘ should be corrected to be formal.
- Under the title 2.5 ‘ Immunofluorescence’, the first sentence is incorrect and should be corrected.
- Under the title 2.6 ‘Flow Cytometry’, the first sentence is missing verb, and also this part has informal sentences,too.
- Under titles 2.7, 2.11, 2.12, 2.14, there are a lot of informal and missing verb sentences.
Although the study is original and used techniques are appropriate, there is one point to say: they say that AS-IV inhibits the activity of PI3K/AKT Pathway according to the western results for AKT protein. However, in order to say that AS-IV inhibitis the PI3K/AKT activity, one should also analyze the levels of Phospho forms of AKT (p-AKT), especially from ser473 and Thr308 residues, because phosphorylation from both sides is a requirement for AKT to be active. So only if we show that p-AKT levels is decreased, we can say PI3K/AKT signaling activity is inhibited. So they should analyze p-AKT levels, or they should change their discussion about this part.
Author Response
Dear Reviewers,
Thank you for taking the time to review this manuscript. We have carefully considered your suggestion and made some changes. We have tried our best to improve and made some changes in the manuscript. Please find my itemized responses below and my revisions in the re-submitted files.
Yours sincerely,
Dr.Qiu

Reviewer 2 Report
Major comments:
1. The authors treated the ULMs with AS-IV, and determined the IC50 for ULMs, but there is no control whether the AS-IV impaired the cell viability of health cells? Is the IC50 of AS-IV inhibits the cell proliferation of health cells?
2. The authors declared that IDO1 mediated the suppressive effect of AS-IV in ULMs cells, but there is no direct evidence.
3. Figure 3 - 5 indicates that IDO1 and AS-IV inhibit cell viability and proliferation of ULMs cells and promote cell apoptosis along, but cell viability and proliferation were increased and apoptosis was decreased in combined treatment with shIDO1 and AS-IV, is it possible? If yes, what’s the reason?
Author Response

(The authors gave the same response as above.)

Round 2
Reviewer 2 Report
the authors addressed most of the questions, just suggest that add some sentences to expain why cell viability and proliferation were increased and apoptosis was decreased in combined treatment with shIDO1 and AS-IV in the discussion section.
Author Response
Dear Editors and Reviewers,
Thank you for taking the time to review this manuscript carefully. According to your advice, we reviewed the literature related to the study using the PubMed database and carefully discussed the reasons for the results. We have tried our best to improve and made some changes in the manuscript. Please find my itemized responses below and my revisions in the re-submitted files.
